# A Novel Sphingosine Kinase Inhibitor Suppresses Chikungunya Virus Infection

**DOI:** 10.3390/v14061123

**Published:** 2022-05-24

**Authors:** Opeoluwa O. Oyewole, Kyle Dunnavant, Shaurav Bhattarai, Yugesh Kharel, Kevin R. Lynch, Webster L. Santos, St. Patrick Reid

**Affiliations:** 1Department of Pathology and Microbiology, College of Medicine, University of Nebraska Medical Center, Omaha, NE 68198, USA; opeoluwa.oyewole@unmc.edu; 2Department of Chemistry, Virginia Tech, Blacksburg, VA 24061, USA; kdunnavant@vt.edu (K.D.); santosw@vt.edu (W.L.S.); 3Department of Cellular and Integrative Physiology, College of Medicine, University of Nebraska Medical Center, Omaha, NE 68198, USA; shbhattarai@unmc.edu; 4Department of Pharmacology, University of Virginia, Charlottesville, VA 22908, USA; yk2n@virgina.edu (Y.K.); krl2z@virginia.edu (K.R.L.)

**Keywords:** Sphingosine kinase 2, Chikungunya virus, SLL3071511 inhibitor, antiviral, drug discovery

## Abstract

Chikungunya virus (CHIKV) is a re-emerging arbovirus in the alphavirus genus. Upon infection, it can cause severe joint pain that can last years in some patients, significantly affecting their quality of life. Currently, there are no vaccines or anti-viral therapies available against CHIKV. Its spread to the Americas from the eastern continents has substantially increased the count of the infected by millions. Thus, there is an urgent need to identify therapeutic targets for CHIKV treatment. A potential point of intervention is the sphingosine-1-phosphate (S1P) pathway. Conversion of sphingosine to S1P is catalyzed by Sphingosine kinases (SKs), which we previously showed to be crucial pro-viral host factor during CHIKV infection. In this study, we screened inhibitors of SKs and identified a novel potent inhibitor of CHIKV infection—SLL3071511. We showed that the pre-treatment of cells with SLL3071511 in vitro effectively inhibited CHIKV infection with an EC_50_ value of 2.91 µM under both prophylactic and therapeutic modes, significantly decreasing the viral gene expression and release of viral particles. Our studies suggest that targeting SKs is a viable approach for controlling CHIKV replication.

## 1. Introduction

Chikungunya virus (CHIKV) is a mosquito-borne virus that causes fever, rashes, and joint pain in infected patients. During acute infection, joint pain can be very severe; importantly, a subset of those infected (up to 40% in some outbreaks) experience debilitating chronic joint pain for months, sometimes up to years [1,2,3,4]. First described in Tanzania in 1952, the virus is currently endemic in countries in Africa, and Asia; since 2007, there have been reports of autochthonous infections in Europe and in 2013, the Americas [5,6,7,8]. This occurrence has been attributed to increased travel and the expansion in the viral transmission range due to mutations that allowed CHIKV to gain a new vector, *Aedes albopictus* mosquitoes, which are found in temperate regions [9]. Given the geographical expansion and increasing incidence of Chikungunya infections, along with the lack of effective therapeutics, an in-depth understanding of the virus–host interactions is critical in identifying therapeutic targets.

Sphingosine kinases (SKs) (i.e., SK1 and SK2) catalyze the phosphorylation of sphingosine, a lipid component of cellular membranes, to sphingosine-1-phosphate (S1P), a bioactive lipid that has been implicated in numerous cellular processes and disease states [10,11,12,13,14,15]. The two sphingosine kinase isoforms are distinct in their localization and downstream effects, although they similarly phosphorylate sphingosine [16,17]. The outcome of S1P formation depends on location—S1P formed at the membrane by SK1 activity is transported out of the cell for binding to its cognate GPCR receptors (S1PR1-5) to carry out autocrine and paracrine functions [18,19]. On the other hand, SK2 performs its function in different subcellular compartments including the endoplasmic reticulum, nucleus, mitochondria, and cytoplasm [20,21]. In the nucleus, S1P inhibits HDAC1/2 to control gene transcription, while mitochondrial S1P formed by SK2 results in the activation of apoptotic processes [22,23,24].

Sphingolipid metabolism has been implicated in the pathology of several diseases including cancer, arthritis, multiple sclerosis, and other inflammatory diseases [25,26,27,28,29]. In addition, SK1 and SK2 have been linked to both pro- and anti-viral effects in the life cycles of several viruses [30,31,32,33,34,35]. We have previously shown that SK2 is a host factor that is important during CHIKV infection. We observed SK2 colocalization with double-stranded RNA and viral nonstructural protein 2 (nsP2) during infection. Furthermore, the inhibition of SK2, but not SK1, resulted in decreased viral infection in vitro [36]. These results suggest that SK2 is a critical pro-viral factor during CHIKV infection, thus targeting SK2 could be a viable avenue for virus inhibition.

Although the structure of SK2 has not yet been fully resolved, modification of the current SK1 inhibitors has been used as a strategy, leading to increased understanding of the structure of SK2 and allowing for the synthesis of more selective inhibitors against SK2 [37,38,39,40,41,42]. In this study, we screened inhibitors that were optimized for their pharmacokinetic properties and inhibitory effect against both SK1 and SK2 for their activity against CHIKV. Our screen identified a novel potent inhibitor of CHIKV infection, SLL3071511, which is a dual SK1 and SK2 inhibitor. Our data show that SLL3071511 is potent against CHIKV infection during early infection, decreasing the infectious virus production, and significantly decreasing the viral RNA expression.

## 2. Materials and Methods

### 2.1. Cell Culture, Compounds, and Virus

The HeLa cells were purchased from American Type Culture Collection (ATCC, Manassas, VA, USA). The cells were cultured in Dulbecco’s modified Eagle’s medium (DMEM) (Life Technologies, Carlsbad, CA, USA) supplemented with 10% inactivated fetal bovine serum (FBS) (GE Healthcare, Chicago, IL, USA). Sphingosine kinase inhibitors were synthesized by the Santos lab (Figure 1A and Appendix A) and reconstituted in DMSO (Alfa Aesar, ThermoFisher Scientific, Waltham, MA, USA). CHIKV (clone 181/25) was purchased from BEI Resources (Manassas, VA, USA).

### 2.2. Sphingosine Kinase Inhibitor Studies

A total of 10,000 HeLa cells were plated per well of a 96-well plate and incubated overnight at 37 °C. The cells were treated with the designated concentrations of inhibitors at the indicated times and infected with CHIKV (MOI = 1 or 10) for 24 h (Figure 1, Figure 2 and Figure 3, Appendix A).

### 2.3. Immunofluorescence

The cells were fixed in 4% PFA-PBS (Thermo Scientific, Waltham, MA, USA) for 15 min and permeabilized with 0.1% Triton-X-PBS (Fisher Scientific, USA) solution for 10 min at room temperature. Blocking was carried out with 3% BSA-PBS (Lee Biosolutions, Maryland Heights, MO, USA) for 1 h. Viral infection was detected using primary antibody grown in rabbit against CHIKV nonstructural protein 3 (nsP3) (1:1000; Dr. Andres Merits) or mouse anti-E1 (viral structural protein) (1:500; BEI Resources, Manassas, VA, USA) incubated overnight at 4 °C. Cells were then stained with secondary antibody, Alexa 488-conjugated goat anti-rabbit (1:2000, ThermoFisher Scientific, Waltham, MA, USA), for 1 h at room temperature. Cell nuclei and cytoplasm were labeled with Hoechst 33342 and the CellMask Deep Red Plasma membrane stain (Invitrogen), respectively. High-content quantitative imaging data were acquired on an Operetta CLS High Content Analysis System (PerkinElmer, Waltham MA, USA). Image analysis was accomplished using Harmony High Content Analysis software. Percentage inhibition was calculated using mean percentage infection from the infected and untreated cells as a positive control (100% inhibition). 

### 2.4. Quantitative RT-PCR 

The HeLa cells were seeded in 6-well plates at a density of 300,000 cells per well and incubated overnight at 37 °C. Cells were then mock treated (DMSO) or treated with SLL3071511 at the indicated concentrations (Figure 3). Compounds were aspirated, followed by infection with CHIKV (MOI = 1) for 24 h. Cells were pelleted and lysed in RNA Lysis Buffer (Zymo Research, Irvine, CA, USA). RNA isolation was carried out using the Quick RNA Miniprep Kit (Zymo Research). RNA was quantified using Nanodrop One (ThermoFisher Scientific, Waltham, MA, USA) and lysates were stored at −80 °C. A total of 500 ng of total RNA was reverse transcribed on an Eppendorf Mastercycler using the qScript cDNA Synthesis Kit (Quantabio, Beverly, MA, USA) with the following program: 22 °C for 5 min, 42 °C for 30 min, 85 °C for 5 min, and kept at 4 °C until storage at −80 °C. SsoAdvanced Universal SYBR Green Supermix (BioRad, Hercules, CA, USA) was used for real-time PCR. A sample of 5 μL of diluted cDNA was used for the qPCR reaction with the following program: (i) PCR activation step, 95 °C for 3 min, 1 cycle; (ii) two-step cycling, 95 °C for 15 s, 60 °C for 1 min, 40 cycles; and (iii) melt curve step, 95 °C for 15 s, 60 °C for 1 min and 95 °C for 15 s. Primers used for the study were *nsP1 Fwd: 5′-GGGCTATTCTCTAAACCGTTGGT-3′; nsP1 Rev: 5′-CTCCCGGCCTATTATCCCAAT-3′; SK2 Fwd:*
*5**′-CTGTCTGCTCCGAGGACTGC-3**′; SK2 Rev: 5**′-CAAAGGGATTGACCAATAGAAGC-3′; GAPDH Fwd: 5′-CCGCATCTTCTTTTGCGTCG-3′ GAPDH Rev: 5′-CCCAATACGACCAAATCCGTT-3′ (IDT, Coralville, IA, USA).* 2^−^^ΔΔCt^ was used to calculate the fold change in expression of the gene of interest, normalized to GAPDH. 

### 2.5. Plaque Assay 

Vero E6 cells (300,000 cells per well) were seeded in 12-well plates and incubated overnight at 37 °C. Five ten-fold dilutions of supernatants from the treated-and-infected wells were prepared in OPTI-MEM (Gibco). A total of 200 μL of the dilutions was added to the cells and incubated at 37 °C for 1 h with shaking every 15 min. Cells were then overlaid with a 1:1 5% FBS + MEM and 2% low melting point agarose mixture (IBI Scientific, Dubuque, IA, USA) and incubated at 37 °C for 72 h. Plaques were fixed with 4% PFA (Thermo Scientific, Waltham, MA, USA) overnight at 4 °C and stained and visualized with crystal violet solution (Sigma-Aldrich, St. Louis, MO, USA). 

### 2.6. Statistics

All statistical analyses were carried out using GraphPad Prism 9.0 software. Data were represented as mean ± standard deviation (SD). Significant differences between the control and experimental groups were calculated using the Student’s *t*-test with *p*-values < 0.05 considered as significant and indicated with an asterisk (*). 

## 3. Results

### 3.1. Screening of SK Inhibitors

As we have previously determined sphingosine kinase importance during CHIKV infection, we investigated the effects of a library of sphingosine kinase inhibitors for their activity during CHIKV infection (Figure 1) [39,40,42,43,44,45]. To assay for their activity against CHIKV, we pre-treated the HeLa cells with the compounds (20 µM) for one hour prior to infection with CHIKV (MOI = 1) for 24 h. We then used the immunofluorescence assay to detect the viral protein (nsP3) expression, followed by the measurement of nsP3 expression as a read-out of anti-CHIKV activity. As shown in Figure 1C, SLC4091425 (Compound #3), SLC5111312 (Compound #8), SLL3041783 (Compound #9), and SLL3071511 (Compound #10) significantly inhibited CHIKV infection at 20 µM (Figure 1B). A cutoff of at least 50% of inhibition of CHIKV infection with at least 75% cell viability was set to determine the compounds of interest.

To confirm the potency of each of the positive SK inhibitors with anti-CHIKV activity, the HeLa cells were treated with increasing concentrations of each of the inhibitors (5, 10, 15, 20 µM) for one hour and then infected with CHIKV (MOI 1) for 24 h. Each of the inhibitors significantly inhibited virus infection with minimal cell cytotoxicity (Figure 2B–E). These data confirmed that the four selected SK compounds were also inhibitors of CHIKV infection at lower concentrations, albeit to varying degrees.

### 3.2. Dose–Response and Time-Course of SLL3071511 Treatment

To validate the activity of SK inhibitors, we treated the Hela cells with an increasing concentration of each inhibitor and determined the corresponding effective constant (EC_50_) of the top four compounds following CHIKV infection. As shown in Table 1, SLL3071511 was more potent than SLC4091425 and SLL3041783 and approximately three times more potent than SLC5111312 with an EC_50_ of 2.9 μM. In a yeast assay of SK isoform selectivity, SLC5111312 was 2-fold more potent at SK1 than SK2, while SLC4091425 and SLL3071511 are dual SK inhibitors [46]. Both SLC4091425 and SLL3041783 would be great candidates for further study. The virus-free cell viability assay also demonstrated that SLL3071511 was non-toxic up to 50 µM. Based on K_i_ data, SLL3071511 prefers SK1 (K_i_ = 0.27 µM) and is also a potent inhibitor of SK2 (K_i_ = 0.66 µM). The concentrations at which it is an effective inhibitor of CHIKV infection are well above both concentrations. Furthermore, SLL3071511 retained significant anti-CHIKV activity even when the cells were infected at higher MOI (MOI = 10). While the elevated virus-induced CPE was observed in infected-only cells at higher MOI, compound-treated cells survived, with higher concentrations having higher survival (Appendix A). Considering the combined potent anti-CHIKV activity, strong inhibitory activity against both kinases, and non-toxicity to normal cells, we chose to further investigate SLL3071511. 

To determine the timing of the effective inhibition of viral infection, the HeLa cells were either treated with the SK inhibitors for one hour before infection, treated along with infection, or treated one hour after infection (MOI 1) (Figure 3A). Pre-treatment for one hour was the most effective for CHIKV inhibition as when the cells were treated during or after infection, the effect of the treatment was diminished.

### 3.3. SLL3071511 Effect on CHIKV Gene Expression and Release 

To determine the effect of SK inhibition on the viral gene expression, we pre-treated HeLa cells with SLL3071511 (20 µM) for 1 h, infected them with CHIKV (MOI 1) for 24 h, and performed quantitative reverse transcriptase PCR (qRT-PCR). In agreement with the immunofluorescence data, we observed that viral nsP1 expression was also significantly diminished in cells following pre-treatment with SLL3071511 (Figure 3B). We performed plaque assays to determine the effect of the inhibitor treatment on the infectious virus release 24 h post-infection. We collected the supernatants from cells that had been treated with 25 µM of inhibitor and infected Vero E6 cells, leaving the plaques to form over 72 h. Plaque counts were also significantly diminished following SLL3071511 treatment, dovetailing nicely with our immunofluorescence and qRT-PCR data (Figure 3C). Altogether, these data suggest that SLL3071511 is a potent inhibitor of viral infection that decreases the viral gene expression and infectious virus release, with the highest inhibitory effect occurring when the compound is used prior to infection.

## 4. Discussion

There are currently no approved antiviral drugs for CHIKV infection treatment. Treatment focuses on relieving the symptoms of infection—joint pain, fevers, and rashes. In 2020, Phase III clinical trials for a virus-like particle (VLP) vaccine began, which is promising for infection prevention [47]. Given the number of people who have been infected and the ensuing chronic joint pain, it is important to investigate the host–virus interactions that can be targeted for treatment. Identifying host factors as therapeutic targets mitigates the risk of viral resistance developing due to the mutations generated from virus-focused therapies.

This project focused on characterizing host-based targets for therapeutics during CHIKV infection. We screened 19 compounds that were synthesized with varying selectivity for SK1 and SK2. We characterized one hit, SLL3071511, which potently inhibits both SK1 and SK2 and had high anti-CHIKV activity. We had previously shown from an siRNA kinome screen that SK2, but not SK1, is important during CHIKV infection [36]. In that study, SK2 was localized primarily in the cytoplasm of the infected cells and colocalized with double-stranded RNA and non-structural protein (nsP) 2 early during infection. Inhibition of SK2 via both chemical and genetic methods resulted in decreased virus infection. 

Other viruses have also been shown to alter sphingolipid metabolism, particularly the function and localization of the sphingosine kinases (reviewed in [48]). SK2 expression increased in the cytoplasm following dengue virus (DENV) infection and the kinase was shown to have a pro-apoptotic role in hepatic cells [49]. On the other hand, SK1 levels decreased during DENV infection with a concomitant decrease in apoptosis [30]. SK1 activity was shown to facilitate the synthesis of influenza A virus (IAV) RNAs and proteins, while SK2 expression and activation are also increased during IAV infection, leading to the increased infection of cells [50]. During measles virus (MV) infection, SK1 expression and activation increased, as also observed during hepatitis B virus and respiratory syncytial virus infections [33,51,52]. However, bovine viral diarrhea virus (BVDV) infection decreases SK1 activity, but not its mRNA expression [35]. Our experiments have shown that CHIKV infection does not alter the SK2 protein or gene expression (data not shown), however, localization is altered, leading to increased cytoplasmic expression.

There is considerably more known about the SK1 structure than that of SK2, whose protein structure has not been currently experimentally validated (the AlphaFold program has provided a putative structure). Given this, compounds that selectively inhibit SK2 are currently not readily available and further studies to identify SK2-selective inhibitors are needed. Both SK1 and SK2 have about a 70% sequence similarity, with differences in the length of the N-terminal and the central region being the main differentiator in their sequence [53]. Using the SK1 inhibitor structure, it was shown that the sphingosine binding pocket is deeper in SK2 than in SK1 and that amino acid changes in the pocket (I174 to V304) allow for bulkier side groups to be added to the inhibitor backbone for SK2-selective inhibitors [42]. In our study, however, the compounds that potently inhibited CHIKV without significant cytotoxicity also had activity against SK1. It should be noted, however, that while these compounds are SK1-selective, they also inhibit SK2. The inhibitory constant (K_i_) of SLL3071511 against human SK1 and SK2 was determined to be 0.27 µM and 0.66 µM, respectively, showing the high potency of this inhibitor against both kinases compared, for example, to SLC4091425 and SLL3041783 (Table 1). While both SLC4091425 and SLL3041783 were not further studied, it stands to reason that both compounds will have a similar course of action against CHIKV and each warrants further study.

Dose–response studies with SLL3071511 against CHIKV showed a potent EC_50_ of 2.9 µM with the maximal inhibition occurring when SLL3071511 is added prior to infection, suggesting that the effect of SK2 occurs early during infection. These results agree with previous work that showed the pretreatment of cells with the SK2 inhibitor, ABC294640, was the most effective at inhibiting CHIKV infection [36], showing that SK2 was important during CHIKV infection. SLL3071511 treatment also resulted in decreased viral gene expression that correlated with the decreased plaque numbers 24 h post-infection. 

One interesting effect of the inhibitor treatment and CHIKV infection was “increased” cell viability when compared to the DMSO-treated or infected-only cells. Some previous studies have shown that overexpression of the kinases, especially SK1, leads to increased cell survival due to increased S1P production and decreased ceramide concentration, especially in cancer cells [14,54,55,56]. Other studies have observed no change in cell viability following the inhibition of the SphKs [57,58]. When compared to the mock-infected cells, lower concentrations of the compound exhibited virus-induced CPE, but the 20 µM and 25 µM treatments both had cell viability values comparable to the mock-infected cells. Viability seemed to increase with higher inhibitor concentrations, and this effect was more pronounced when cells were infected at higher MOI (MOI = 10). We do not believe that this is an indication of the increased proliferation of the cells, rather, we believe that the inhibitor treatment decreased the virus-induced cytopathic effect, leading to the increased survival of cells even with high infection titers (Appendix A). 

A limitation of this study is that we did not measure the changes in the sphingosine or S1P expression that could contribute to the effect of the compound on CHIKV infection. SLL3071511 is a competitive inhibitor of the kinases and functions to decrease S1P and increase sphingosine expression in the cells as it inhibits the kinase by binding to the substrate-binding site [45]. Further studies are needed to better understand the mechanism of the CHIKV inhibition, whether it is a direct function of the kinase during CHIKV infection or the result of changes in sphingosine expression following the inhibitor treatment in cells. The early effect of SLL3071511 on CHIKV infection and the similar inhibition of both non-structural and structural proteins suggests that early stages of infection may be the target (Figure 3 and Appendix A). Advanced techniques such as live-cell imaging would be useful in determining the interactions of the SKs with the CHIKV replication complex and provide more insights into the role of the kinases during infection. Furthermore, the effect of SLL3071511 on viral entry can be determined using pseudo-typed viruses. Structural associations of the viral replication complex with the sphingosine kinases would also be useful in determining the specific role of SKs during viral infections. Taken together, the data show that the sphingosine kinases may be viable targets for the inhibition of CHIKV infection, and that further optimization of SK inhibitors will allow for more potent viral inhibition.

## Figures and Tables

**Figure 1 viruses-14-01123-f001:**
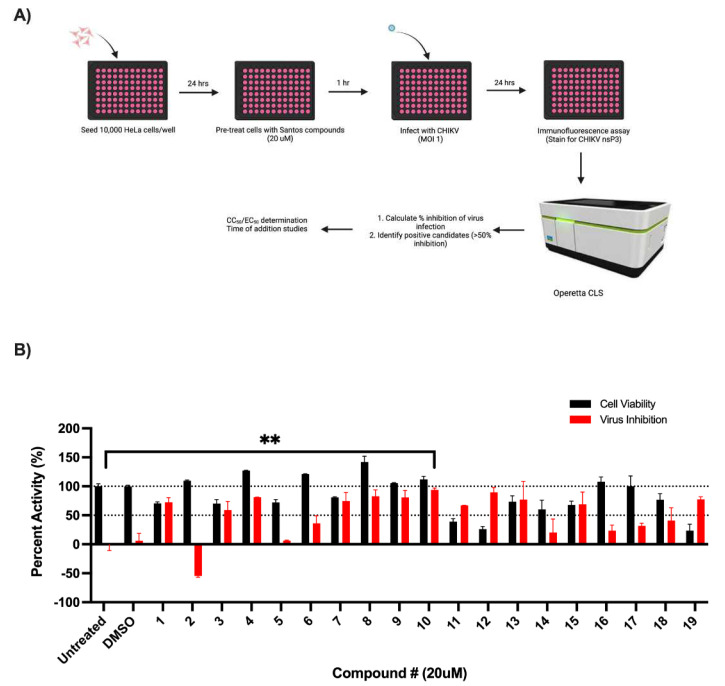
**The screening and identification of potent sphingosine kinase (SK) inhibitors for activity against CHIKV infection.** (**A**) Method schema used to identify potent CHIKV inhibitors. Cells were treated with SK inhibitors (20 µM) for 1 h followed by mock-infection or infection with CHIKV at an MOI of 1. At 24 hpi, percent cell viability and the inhibition of infection were measured using an immunofluorescence assay after staining for viral protein, nsP3. (**B**) Screening for SK inhibitors with inhibitory activity against CHIKV. HeLa cells were pre-treated with inhibitors (20 µM) for 1 h and then infected with CHIKV (MOI = 1) for 24 h. Staining was conducted for viral nonstructural protein (nsP3, green), nucleus (Hoechst 33342, blue), and cytoplasm (CellMask, red). Images were obtained on an Operetta CLS confocal microscope (20× objective), and analysis was conducted using the Harmony software. Data represent the means (±SD) from one representative experiment of at least two independent experiments performed in duplicate. Statistical analysis was determined using unpaired Student’s t-tests, and significant differences are indicated by ** (*p* < 0.005).

**Figure 2 viruses-14-01123-f002:**
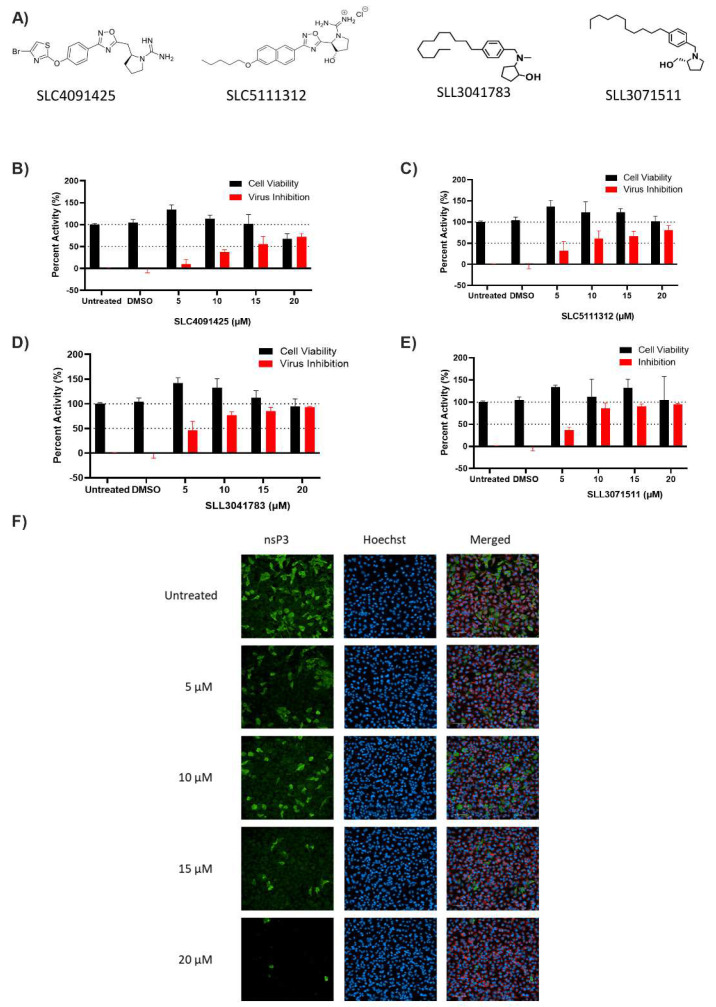
**The dose–response of potent sphingosine kinase inhibitors for activity against CHIKV infection.** (**A**) The chemical structures of the sphingosine kinase inhibitors positive for anti-CHIKV activity. (**B**–**E**) Positive candidates were further analyzed using a dose-response curve (5, 10, 15, 20 µM). HeLa cells were pre-treated for 1 h and then infected with CHIKV (MOI = 1) for 24 h. Staining was conducted for viral nonstructural protein (nsP3, green), nucleus (Hoechst33342, blue), and cytoplasm (CellMask, red). Images were obtained on the Operetta CLS confocal microscope (20× objective), and analysis was conducted using the Harmony software. Data represent means (± SD) from one representative experiment of at least two independent experiments performed in duplicate. (**F**) Representative images of the SLL3071511 inhibition of CHIKV infection were collected using the Operetta CLS microscope (20× objective).

**Figure 3 viruses-14-01123-f003:**
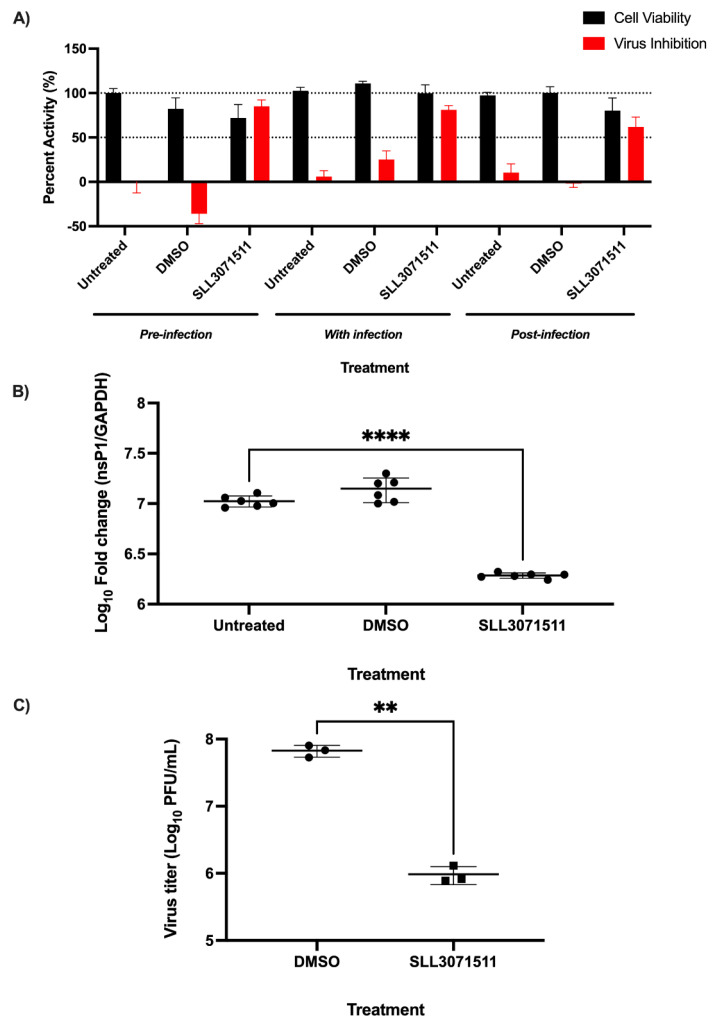
**Early treatment with SLL3071511 significantly decreases viral RNA production virus release following infection.** (**A**) HeLa cells were mock-infected or infected with CHIKV (MOI = 1) either 1 h post-treatment, concurrent with treatment or 1 h pretreatment with DMSO or SLL3071511. At 24 hpi, staining was carried out for the viral nonstructural protein (nsP3, green), cell nucleus (Hoechst33342, blue), and cell cytoplasm (CellMask, red). Images were obtained on the Operetta CLS confocal microscope (20× objective) and analysis was conducted using the Harmony software. (**B**) HeLa cells were mock-infected or infected with CHIKV at an MOI of 1 following a 1-h pretreatment with DMSO or SLL3071511 (20 µM). At 24 hpi, cells were harvested, and RNA was isolated, followed by cDNA synthesis. qPCR was conducted and log fold change of the nsP1 expression was calculated using the 2^−ΔΔCt^ method with GAPDH as the housekeeping gene. (**C**) HeLa cells were mock-infected or infected with CHIKV at an MOI of 1 following a 1-h pretreatment with DMSO or SLL3071511 (25 µM). At 24 hpi, plaque assay was carried out using supernatant from the samples. Data represent the means (±SD) from one representative experiment of at least two independent experiments performed in duplicate. Statistical analyses were determined using unpaired Student’s *t*-tests, and significant differences are indicated by ** (*p* < 0.005), **** (*p* < 0.0001).

**Table 1 viruses-14-01123-t001:** The activity of the positive CHIKV inhibitor candidates against SK1 and SK2 and CHIKV.

Compound ID	SK1 K_i_ (µM)	SK2 K_i_ (µM)	EC_50_ (µM)
SLC4091425	> 3	>3	3.35
SLC5111312	0.36 ± 0.09	0.33 ± 0.07	11.84
SLL3041783	2.4 ± 0.61	>3	3.33
SLL3071511	0.27 ± 0.03	0.66 ± 0.11	2.91

K_i_ = inhibitor constant; EC_50_ = half-maximal effective concentration.

## Data Availability

Not applicable.

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
