# Peer review of "A Novel Sphingosine Kinase Inhibitor Suppresses Chikungunya Virus Infection"

_viruses, 2022, doi:10.3390/v14061123_

Round 1

Reviewer 1 Report

The authors test of series of sphingosine kinase inhibitors on the ability to control Chikungunya virus replication. The methods are standard and results look reasonable. However, there is some missing information and a decisions that could be better explained.

The paper seems to be the second half of a medicinal chemistry paper. This is fine but it is not clear to me which is the first half. The only information about the compound structures is “Sphingosine kinase inhibitors were synthesized by the Santos lab (Fig. 1A)…” in the Materials and Methods section. Fig. 2A only shows 4 fo the 19 tested compounds. The rest could be anything. The rest of the compound structures should be shown, possibly in SI, with a citation to a paper with synthetic methods and NMR.

p. 7 line 207 mentions that ABC294640 was a control, but I do not see the data. On p. 10 line 291, differences between the current study and Ref 36 should be more clearly highlighted to clarify why it previously showed antiviral activity but does not do so in the current study.

The rationale for choosing SLL3071511 was unclear (p. 7 lines 196-198). Fig. 2 shows that SLC5111312 and SLL3041783 are equally nontoxic. SLL3071511 appears very hydrophobic which could affect bioavailability. SLC5111312 does not look as hydrophobic. It is also unclear why SK1 inhibition is preferable if SK2 was previously shown to be a host factor important during CHIKV infection.

Author Response

Manuscript ID: viruses-1685426

Title: A Novel Sphingosine Kinase Inhibitor Suppresses Chikungunya Virus Infection

Reviewer 1:

Comments and Suggestions for Authors

The authors test of series of sphingosine kinase inhibitors on the ability to control Chikungunya virus replication. The methods are standard, and results look reasonable. However, there is some missing information and a decision that could be better explained.

The paper seems to be the second half of a medicinal chemistry paper. This is fine but it is not clear to me which is the first half. The only information about the compound structures is “Sphingosine kinase inhibitors were synthesized by the Santos lab (Fig. 1A)…” in the Materials and Methods section. Fig. 2A only shows 4 of the 19 tested compounds. The rest could be anything. The rest of the compound structures should be shown, possibly in SI, with a citation to a paper with synthetic methods and NMR.

  • Response: We appreciate the reviewer for pointing out this omission. The compound structures and citations have been added as Supplementary Figure 1.
  1. 7 line 207 mentions that ABC294640 was a control, but I do not see the data. On p. 10 line 291, differences between the current study and Ref 36 should be more clearly highlighted to clarify why it previously showed antiviral activity but does not do so in the current study.
  • Response: We thank the reviewer for asking for this clarification. We have taken out reference to ABC294640 as inhibition was also previously not as robust as SLL3071511 and was used as a proof of concept for SK2 being important for CHIKV infection. SKI-II, which is another SK inhibitor, was effective at inhibiting CHIKV in our study, providing evidence that SK inhibition is effective at inhibiting the virus.

The rationale for choosing SLL3071511 was unclear (p. 7 lines 196-198). Fig. 2 shows that SLC5111312 and SLL3041783 are equally nontoxic. SLL3071511 appears very hydrophobic which could affect bioavailability. SLC5111312 does not look as hydrophobic. It is also unclear why SK1 inhibition is preferable if SK2 was previously shown to be a host factor important during CHIKV infection.

  • Response: We thank the reviewer for the insight regarding the potential oral bioavailability of SLL3071511 and SLC5111312. Fortunately, for the in vitro studies, cell penetration has not been an issue as the compounds are soluble in the assay conditions and they are cationic. Indeed, both compounds showed antiviral activity in our assay. Oral bioavailability is beyond the scope of the current work. As for inhibitor preference for SK1 versus SK2, we would like to highlight that the concentration at which SLL3071511 is effective as a CHIKV inhibitor is above the Ki for both SK1 and SK2, therefore while SK1 is inhibited, we believe SK2 inhibition also is contributing to the viral inhibitory effect observed. Finally, since the EC50 calculated for SLL3071511 is much lower than that for SLC5111312, we decided to further study SLL3071511. We have also updated the manuscript to highlight our rationale for choosing SLL3071511. We have taken out reference to ABC294640, as it does not add to the manuscript, and we are unsure how to rationalize the data we have obtained. ABC294640 inhibition was also previously not as robust as SLL3071511 and was used as a proof of concept for SK2 being important for CHIKV infection.

Reviewer 2 Report

Manuscript ID: viruses-1685426

Title: A Novel Sphingosine Kinase Inhibitor Suppresses Chikungunya Virus Infection

The authors analyzed the intervention with CHIKV by inhibition of the sphingosine-1-phosphate (S1P) pathway, which converts sphingosine to S1P by Sphingosine kinases (SKs).

The authors screened 19 SKs inhibitors and identified a potent inhibitor of CHIKV infection -SLL3071511, which worked best as pretreatment. However, the data need more details on the mode of action.

Comments:

  • Please show data that the inhibitor is specific for SK1/2.
  • The screening of inhibitors was done by analyses of the viral nonstructural protein-3 (nsP3), which is translated from the incoming viral genomic RNA (gRNA). Inhibition of translation could be an explanation for this effect. The authors should analyze structural protein synthesis, which is after the initial translation of the gRNA.
  • In Figure 2, not all untreated cells are infected. Is the inhibitor still functional with a high MOI infection?
  • Cell viability is increasing with higher amounts of drugs. Why is this? Are the drugs inducing cell proliferation?
  • Figure 3: Figure legend of Figure 3 does not fit to the data, time of drug addition is missing.
  • Inhibition of virus release is rather low with 0.5 log decrease. This needs an explanation. The experiment was performed with25 mM compound. Toxicity data are missing.
  • An important point is the mechanism of action. The author discuss that most likely CHIKV entry is inhibited. This could be shown with pseudotyped vectors. Please show data.
  • If SKs are involved in many cellular processes, why are the compounds not toxic? Please discuss SK knock-out mice.

Author Response

Manuscript ID: viruses-1685426

Title: A Novel Sphingosine Kinase Inhibitor Suppresses Chikungunya Virus Infection

Reviewer 2:

The authors analyzed the intervention with CHIKV by inhibition of the sphingosine-1-phosphate (S1P) pathway, which converts sphingosine to S1P by Sphingosine kinases (SKs).

The authors screened 19 SKs inhibitors and identified a potent inhibitor of CHIKV infection -SLL3071511, which worked best as pretreatment. However, the data need more details on the mode of action.

Comments:

  • Please show data that the inhibitor is specific for SK1/2.
    • Response: We thank the reviewer for the comment. The supplemental information section of the manuscript has been updated with information about the selectivity of the compounds and the corresponding references.
  • The screening of inhibitors was done by analyses of the viral nonstructural protein-3 (nsP3), which is translated from the incoming viral genomic RNA (gRNA). Inhibition of translation could be an explanation for this effect. The authors should analyze structural protein synthesis, which is after the initial translation of the gRNA.
    • Response: We appreciate the reviewer pointing out this gap. We repeated the treatment and infection experiment and stained for viral glycoprotein which is translated after viral replication and observed the same inhibition of expression at 24 hours post infection (Supplementary Figure 3).
  • In Figure 2, not all untreated cells are infected. Is the inhibitor still functional with a high MOI infection?
    • Response: We thank the reviewer for this comment. We have updated the manuscript with Supplementary Figure 2 which shows that at MOI 10, the inhibitor is still functional at CHIKV inhibition.
  • Cell viability is increasing with higher amounts of drugs. Why is this? Are the drugs inducing cell proliferation?
    • Response: We thank the reviewer for this observation, and we have updated the manuscript with this discussion. Based on our data, we do not believe that the drugs are inducing cell proliferation, as the number of cells only changes when compared to infected-only cells. Cell numbers in treated and infected cells are similar to that observed in mock-infected cells. Therefore, we think that the compound is likely promoting the survival of the cells by inhibiting virus-induced cell death.
  • Figure 3: Figure legend of Figure 3 does not fit to the data, time of drug addition is missing.
    • Response: We thank the reviewer for bringing this error to our attention. The figure legend has been corrected to reflect the data shown in the figure.
  • Inhibition of virus release is rather low with 0.5 log decrease. This needs an explanation. The experiment was performed with 25 mM compound. Toxicity data are missing.
    • Response: We appreciate the reviewer’s comment about the toxicity of the inhibitors at 25 µM. The toxicity data for the experiment is the same as shown in Figure 3A as the experiment for viral titer determination was also done with pre-treatment of cells for 1 hour before infection with CHIKV. As for inhibition of viral release, repeat of the experiments showed that there is decreased viral release 24 hours following treatment at 25 µM and infection at MOI = 1.
  • An important point is the mechanism of action. The author discusses that most likely CHIKV entry is inhibited. This could be shown with pseudotyped vectors. Please show data.
    • Response: The authors thank the reviewer for this comment. We agree that pseudotyped vectors would be ideal for determining the mechanism of action. Unfortunately, we were unable to perform experiments with pseudotyped vectors in the limited time we had for revisions. We have revised the manuscript and removed the reference to inhibition of entry as the mechanism of action. Furthermore, when we measured nsP1 expression 4 hours after synchronizing CHIKV infection for 1 hour at 4C, we observed no difference in the expression of viral RNA, suggesting that entry is not inhibited by the compound.
  • If SKs are involved in many cellular processes, why are the compounds not toxic? Please discuss SK knock-out mice.
    • Response: We thank the reviewer for this question and have updated the manuscript to reflect our response. In brief, while the SKs are important for cellular processes, studies have shown that inhibition of either kinase (especially SK1) does not affect cell viability. In mice, knocking out either SK was not lethal, while knockout of both SK1 and SK2 was embryonic lethal (E13.5) as S1P is essential during embryogenesis. Furthermore, in our study, infection with CHIKV maintains the viability of cells treated with the compound, as virus-induced cytopathic effect was inhibited. Our CC50 studies also showed that the compounds were also not cytotoxic up to 50 µM which is higher than the concentrations used in our study.

Round 2

Reviewer 2 Report

Manuscript ID: viruses-1685426

Revised version

Title: A Novel Sphingosine Kinase Inhibitor Suppresses Chikungunya Virus Infection

  1. Numbering of supplement files is apparently wrong.

  1. Question of reviewer: Cell viability is increasing with higher amounts of drugs. Why is this? Are the drugs inducing cell proliferation?

Answer of author:

“Viability seemed to increase with higher inhibitor concentrations, and this effect was more pronounced when cells were infected at higher MOI (MOI = 10). We do not believe that this is an indication of increased proliferation of the cells, rather, we believe that inhibitor treatment abrogated virus-induced cytopathic effect, leading to increased survival of cells even with high infection titers (Supplementary Figure 2A). The percent viability of treated and infected cells was similar to untreated cells agreeing with previous studies that found no change in cell viability with SK inhibitor treatment of cells (Supplementary Figure 2C) (58).”

Comment:

In Suppl. Figure 2C, are mock infected (not infected) cells treated with compound and viability is increasing, which cannot be accounted to abrogation of virus-induced cytopathic effects. If so, infected cells should show less viability. Please explain.

  1. Sentence lines 351 -357 is missing “and structural proteins”

“The early effect is observed following pre-treatment, we believe that of SLL3071511 might function by inhibiting viral entry into cells. As the on CHIKV inhibition was sustained up to 24 hours post infection, we also think that the compound might be inhibiting CHIKV and the similar effect on both non-structural proteins and structural proteins suggests that replication is the infection step that is inhibited, as viral entry was not altered with treatment (Supplementary Figure 3; data not shown).”

Comment:

Viral entry could still be inhibited, because this would result in the same phenotype only experiments with pseudotypes could answer this question. Please rephrase the sentence.

Author Response

  1. Numbering of supplement files is apparently wrong.

 Response: We apologize for the error and thank the reviewer for bringing it to our attention. The numbering of supplementary files has been corrected.

  1. Question of reviewer: Cell viability is increasing with higher amounts of drugs. Why is this? Are the drugs inducing cell proliferation?

Answer of author:

“Viability seemed to increase with higher inhibitor concentrations, and this effect was more pronounced when cells were infected at higher MOI (MOI = 10). We do not believe that this is an indication of increased proliferation of the cells, rather, we believe that inhibitor treatment abrogated virus-induced cytopathic effect, leading to increased survival of cells even with high infection titers (Supplementary Figure 2A). The percent viability of treated and infected cells was similar to untreated cells agreeing with previous studies that found no change in cell viability with SK inhibitor treatment of cells (Supplementary Figure 2C) (58).”

Comment:

In Suppl. Figure 2C, are mock infected (not infected) cells treated with compound and viability is increasing, which cannot be accounted to abrogation of virus-induced cytopathic effects. If so, infected cells should show less viability. Please explain.

Response: We thank the reviewer for this comment. We apologize for the confusion. We have now added mock control to show the difference between infected and uninfected, reiterating that loss of viability is due to virus-induced CPE. Please see below for the corrected graph, which has also been added to the Supplementary Figure 2.

Lines 336-339: “When compared to mock-infected cells, lower concentrations of the compound exhibited virus-induced CPE, but treatment with 20 µM and 25 µM treatments both had cell viability comparable to mock-infected cells.”

Lines 341-343: “We do not believe that this is an indication of increased proliferation of the cells; rather, we believe that inhibitor treatment decreased virus-induced cytopathic effect, leading to increased survival of cells even with high infection titers (Supplementary Figure 2A).”

  1. Sentence lines 351 -357 is missing “and structural proteins”

“The early effect is observed following pre-treatment, we believe that of SLL3071511 might function by inhibiting viral entry into cells. As the on CHIKV inhibition was sustained up to 24 hours post infection, we also think that the compound might be inhibiting CHIKV and the similar effect on both non-structural proteins and structural proteins suggests that replication is the infection step that is inhibited, as viral entry was not altered with treatment (Supplementary Figure 3; data not shown).”

Comment:

Viral entry could still be inhibited, because this would result in the same phenotype only experiments with pseudotypes could answer this question. Please rephrase the sentence.

Response: We thank the reviewer for this comment, and we have corrected the manuscript to reflect the edits. Please see below:

Lines 351-356: “The early effect of SLL3071511 on CHIKV infection and the similar inhibition of both non-structural and structural proteins suggests that early stages of infection may be the target (Figure 3 and Supplementary Figure 3B).”…

Lines 360-361: “Furthermore, the effect of SLL3071511 on viral entry can be determined using pseudotyped viruses.”
